# Modified Proximal Interphalangeal Joint Arthrodesis in a Yearling Filly with an Osseous Cyst-Like Lesion in the Proximal Phalanx

**DOI:** 10.3390/ani11040948

**Published:** 2021-03-28

**Authors:** Filip Kol’vek, Lenka Krešáková, Katarína Vdoviaková, Ľubomír Medvecký, Zdeněk Žert

**Affiliations:** 1Equine Clinic, University of Veterinary Medicine and Pharmacy in Kosice, Komenskeho 73, 041 81 Kosice, Slovakia; zdenek.zert@uvlf.sk; 2Department of Morphological Disciplines, University of Veterinary Medicine and Pharmacy in Kosice, Komenskeho 73, 041 81 Kosice, Slovakia; lenka.kresakova@uvlf.sk (L.K.); katarina.vdoviakova@uvlf.sk (K.V.); 3Division of Functional and Hybrid Systems, Institute of Materials Research of SAS, Watsonova 47, 040 01 Kosice, Slovakia; lmedvecky@saske.sk

**Keywords:** equine, arthrodesis, proximal interphalangeal joint, osseous cyst-like lesion, phalanx, calcium phosphate biocement

## Abstract

**Simple Summary:**

Subchondral cystic lesions are common in young horses, but they can be diagnosed in older horses and often cause lameness. They are most frequently found in the medial femoral condyle, phalanges, elbow, and carpal bones. Lesions that interact with the joint may cause severe lameness and may not respond well to medical treatment. Arthrodesis, or “artificial ankylosis”, of the affected joint can be an effective radical procedure to reduce pain. As a “low motion” joint, the proximal interphalangeal joint has significantly more mobility than the small hock joints. Fusion is impossible to achieve simply by removing the remaining cartilage. To bridge and immobilize the joint, it must be used in conjunction with the surgical placement of screws and plates. This report describes a successful modified surgical method of arthrodesis of the proximal interphalangeal joint and packing with calcium phosphate biocement in a filly with an osseous cyst-like lesion of the proximal phalanx that communicated with the proximal interphalangeal joint.

**Abstract:**

After the medial femoral condyle (MFC), the phalanges are the second most common site for osseous cyst-like lesions (OCLLs). Conservative treatment of phalangeal cysts on the convex surface of proximal phalanx presents a technical problem with access to the stoma of the cyst. Surgical therapy options usually aim to avoid cyst enlargement through drilling or screw placement or to encourage lesion filling with osteoconductive material. This paper describes a case of treatment of the OCLL in a yearling Czech warmblood filly with surgical arthrodesis, together with the packing of the OCLL with calcium phosphate biocement (CPB). The filly showed a chronic, moderate to severe, intermittent left hindlimb lameness. Dynamic examination combined with regional anesthesia and radiography confirmed a clinically significant large OCLL on the distal joint surface of the first phalanx. Treatment of the OCLL was performed by surgical arthrodesis of the proximal interphalangeal joint, using two paraxial and one axial crossed lag screw, after curetting of the cyst and filling with CPB.

## 1. Introduction

Proximal interphalangeal joint (PIPJ) arthrodesis is indicated in horses with complete skeletal ossification to reduce pain associated with osteoarthritis (high ringbone), comminuted fractures of the proximal or middle phalanx, and luxation or subluxation of the PIPJ [1]. There have been five reports of PIPJ arthrodesis in foals that have been published; the first was performed for a proximal phalangeal fracture, the second for an unknown cause, the third for a congenital malformation of the proximal interphalangeal joint, the fourth for disruption of the superficial digital flexor tendon and straight sesamoidean ligament, and the fifth in foals for severe osteochondrosis and maldevelopment of the PIPJ’s articular surface [2,3,4,5,6]. In contrast to osteochondrosis dissecans, which affects the joint surface, subchondral bone cysts (also known as subchondral cystic lesions or osseous cyst-like lesions) is a type of centrally localized osteochondral lesion found in horses. They are not true bone cysts. Because they are not inside and are not covered by an epithelial lining, and they can interact with the joint, it has been agreed that they should be referred to as osseous cyst-like lesions (OCLLs) [7,8]. In horses, OCLLs may develop in a variety of bones and joints, with the phalanges and digital articulations being the second most common location (26.2%), after the medial femoral condyle (45.8%) [9]. Phalangeal cysts may be treated conservatively or surgically. Controlled exercise, symptomatic nonsteroidal anti-inflammatory medication, and intra-articular medication with chondroprotectants +/− corticosteroids are recommended as a conservative therapy [10]. In general, surgical procedures inhibit cyst enlargement and allow osseous material filling of the lesion. This can be done arthroscopically or via an extra-articular approach, depending on the cyst location. The cyst’s size and grade communicating with the space of the proximal interphalangeal joint may necessitate surgical arthrodesis in some cases [11]. Several techniques are recommended for PIPJ arthrodesis. The first technique routinely applied used two or three transarticular cortical screws placed in lag fashion in a parallel or diverging pattern. However, when compared to plating methods, this approach necessitates more cast support. Other options for improving stability include two plates, a T-plate, a Y-plate, and, most recently, a spoon plate, but the best clinical results have been described with a dorsal three-hole narrow DCP or LCP plate combined with two transarticular abaxial 5.5 mm cortical screws inserted in lag fashion [1]. The prognosis for return to the intended use in horses with OCLLs ranges from 30–90%, depending on the breed, age of the horse, the surface area of the weight-bearing cartilage impacted by concurrent osteoarthritis inside the joint and therapy administered [8,12,13,14,15,16,17].

The purpose of this case report is to describe the clinical findings associated with a case of the osseous cyst-like lesion in the proximal phalanx in a one-year-old filly and the consequent surgical debridement followed by packing with calcium phosphate biocement (CPB) [18], before our original surgical repair using three cortical screws to arthrodese the PIPJ [6].

## 2. Materials and Methods

### 2.1. Case Selection and History

A one-year-old 333 kg Czech-warmblood filly was presented to our hospital with chronic left hind lameness, which had gradually developed over 2 months. The foal had been normal since birth, according to reports, but was seen to be non-weight-bearing lame on the left hindlimb after a spontaneous gallop in the paddock. The referring veterinarian obtained radiographs [dorsoplantar (DPl), lateromedial (LM)] of the distal limb. The radiographs revealed axial subchondral bone radiolucency, and the foal was referred to the clinic for further assessment.

### 2.2. Clinical and Radiographic Examination

Upon presentation, the filly was bright, alert, and responsive, and the physical examination findings were normal. On inspection and palpation, there were not any changes of the left hindlimb. At the walk, the lameness was not observed. A mild (2/5 AAEP grading system) left hindlimb lameness was evident at the trot on a straight line on a hard surface. The flexion test of the digit was positive. Intraarticular diagnostic anesthesia of the proximal interphalangeal joint was positive, confirming the clinical importance of the radiographic findings. Repeated radiographs (DPl, LM, dorsolateral–plantaromedial oblique (DLPlMO) and dorsomedial–plantarolateral oblique (DMPlLO)) confirmed circular axial radiolucency (12–15 mm width, 10 mm depth) within the distal subchondral bone plate and trabecular bone of the proximal phalanx, with a wide communication into the proximal interphalangeal joint (Figure 1). An internal fixation with OCLL packing was done to arthrodese the PIPJ.

### 2.3. Preoperative Management

Pre-/post operatively potentiated sulphonamides (Borgal 24%, Virbac, Carros, France; 20 mg/kg t.i.d.) and flunixin meglumine (Vetaflumex, MultiTrade Compani *Vet-Agro*, Lublin, Poland; 1.1 mg/kg b.i.d.) were given intravenously, and omeprazole (Peptizole, Norbrook Laboratories Ltd., Monaghan, Ireland; 4.0 mg/kg s.i.d.) per os was also administered. The filly was sedated with 1.1 mg/kg of xylazine (Rometar 20 mg/mL, Bioveta, a.s., Ivanovice na Hané, Czech Republic) intravenously, and anesthesia was induced with 0.02 mg/kg of diazepam (Apaurin; Krka, Novo Mesto, Slovenia) and 2.2 mg/kg of ketamine (Ketamidor, Richter Pharma, Wels, Austria) intravenously. Anesthesia was maintained with isoflurane in oxygen (Isoflurin, Vetpharma Animal Health, S.L., Barcelona, Spain) and medetomidine infusion at a constant rate (3.5 μg/kg/h; IV) (Cepetor, CP-Pharma Handelsges, Burgdorf, Germany). Medetomidine was diluted in physiologic saline (0.9% NaCl). A spontaneous respiration regime was maintained for the whole operation time.

### 2.4. Surgical Procedure

With the appropriate preparation and draping of the surgical site, the patient was placed in lateral recumbency, with the affected limb positioned uppermost. A dorsal surgical approach was used, including transection of the skin, the long digital extensor tendon (LDET), and the collateral ligaments of the proximal interphalangeal joint to achieve exposition of the articular surfaces. An “inverted V” skin incision was made with the vertical arms of the V made abaxially, approximately 1 cm above the coronary band and extending proximo-axially to the center of the proximal phalanx [6]. One triangular-shaped skin flap with attached subcutaneous tissues was sharply separated from the underlying LDET and retracted. An “inverted V” incision was then made through the LDET, with the midpoint of the “V” beginning below the fusion of the extensor branches of the suspensory ligament and extensor tendon. To enable visualization of the PIPJ, the distal end of the transected tendon was retracted distally. The joint was opened and disarticulated by sharp transection of both the lateral and medial collateral ligaments of the PIPJ. After complete disarticulation of the PIPJ, a manual and power curettage was used to remove the cartilage from the articular surfaces of the proximal phalanx (P1) and middle phalanx (P2). Using a 3.2 mm drill bit, the subchondral bone plates of P1 and P2 were fenestrated (osteostixis) (Figure 2). This allowed access to the medullary vascular and cellular components, which helped to promote bone healing. Surgical treatment of the OCLL involved removal of the cyst content by curettage of the cyst wall (Figure 2a) and filling the cavity with a CPB to stimulate healing of the bone cavity (Figure 2b), followed by stabilization with cortical screws in our preferred way [6].

An axially positioned transarticular 4.5 mm cortex screw (50 mm length) was placed as a first in lag fashion, with the glide hole in the dorsal proximal aspect of the middle phalanx drilled from the joint surface and through the incision in the long digital extensor tendon, and with the threaded hole in the distal plantar aspect of the proximal phalanx drilled after the reposition. This screw should pass through the articulation near the dorsal ¼ and plantar ¾ junction, ensuring that the screw purchase in the proximal phalanx is within the distal aspect and avoids the coronary area and hoof capsule. This screw will maintain reduction for the remainder of the fixation and provides transarticular compression in the close perpendicular direction to them (Figure 3). In the transarticular, parallel, lag screw fashion, two 4.5 mm cortical screws were positioned second and third. The head of each screw was positioned just proximal to the proximal interphalangeal joint capsule origin, 1 cm abaxial to the axial plane. First, we drilled the glide holes disto-proximally into the proximal aspect of P2 for axial screw insertion and into the distal aspect of P1 for abaxial screw insertion. After preparation of glide holes, the joint was repositioned in a physiological position, and the 4.5 mm cortical self-tapping screw was inserted after the threaded hole for the axial screw was drilled across the bone. The thread holes for two abaxial transarticular screws were drilled, and the screws were inserted and tightened. The lengths of the screws were 48 and 44 mm, respectively. Drilling was done with power tools; tapping was not required because self-tapping screws were used. Countersinking was not performed because of soft dorsal corticalis. The surgical wound was closed in two layers. The extensor tendon ends were apposed and sutured in a simple pattern using absorbable polyfilament suture material (Vicryl, USP1/Metric4, Ethicon, Cincinnati, OH, USA). A simple vertical mattress suture pattern was used to close the subcutaneous and skin, using an absorbable polyfilament suture (Vicryl, USP2/metric5, Ethicon). The surgical wound was covered with an antibiotic mesh and a half-limb fiberglass cast (4 in/10.1 cm Scotchcast^TM^, 3M HealthCare, Saint Paul, MN, USA) from the proximal metatarsus to maintain the distal limb in a neutral position.

The recovery from anesthesia was controlled, and the patient started to use the limb immediately.

This cast was changed two weeks post-surgery. Nonsteroidal and antimicrobial therapy was continued for three weeks. The filly was kept in a box for 3 months and maintained in the bandage cast for an additional 6 weeks. After 6 weeks, given the horse’s clinical and radiographic progress was satisfactory, hand-walking exercises were introduced.

## 3. Results

After three months, including the six weeks of hand-walking exercise, gradual access to free paddock activity was allowed based on the clinical and radiographic progress. The radiographs demonstrated bone bridging laterally and dorsally at the proximal interphalangeal joint. There was smooth, new bone formation dorsally over the middle phalanx at the PIPJ margins and up to and covering the screw heads (Figure 4). The screws stayed in place after surgery.

Six weeks after the operation, the filly was free moving with no lameness in the paddock. The proximal interphalangeal joint region was hard swollen, and the swelling slowly regressed. The screws remained in the bones and are not intended to be removed. Ten weeks postoperatively, the filly was discharged with instructions to control hand-walking and then controlled training exercise. The owner reported that the filly was sound at a walk and trot. Around the dorsal aspect of PIPJ, the pastern area was thickened slightly. Control radiographs were not obtained.

## 4. Discussion

The proximal interphalangeal joint is a diarthrodial, low-motion joint consisting of the distal convex articular surface of the proximal phalanx and the proximal concave articular surface of the middle phalanx. The pathogenesis of osseous cyst-like lesions (OCLLs), as well as treatment options, are still being discussed [19]. The lesions were initially taken as osseous cyst-like lesions. Other authors later referred to them as SCLLs, or osseous cyst-like lesions, to avoid implying that they were true cysts. Young horses with little or no evidence of osteoarthritis, as well as mature horses with clinical significance, may develop subchondral osseous cystic lesions, which can cause lameness. Osseous cyst-like lesions are generally easily detected by X-ray examination on both lateral and DP views. They most often occur unilaterally in the distal aspect of the proximal phalanx, and they are most common in the hindlimbs [20]. In horses under one year of age, spontaneous regression is possible; however, they can be discovered as a cause of clinical problems in juvenile horses with normal radiographs at six months of age [21]. A cyst that has separated from the joint space is usually of minor clinical relevance and is often overlooked. Large and multiple cystic defects in horses often result in osteoarthritis and are more likely to cause lameness [11,22]. Younger horses are more likely to develop subchondral cystic lesions of the proximal interphalangeal joint, often followed by thickening the pastern (high ringbone). Lameness could be present, and in some cases, could be permanent and serious, but swelling around the pastern region necessitates additional diagnostic steps. Lesions are usually discovered on routine pre-purchase examinations in horses without swelling or clinical symptoms. Small cystic lesions in yearlings may resolve or cause few complications, but some may progress to clinically relevant disease. Subchondral cystic lesions on the PIPJ are difficult to treat arthroscopically; in most cases, a proximal interphalangeal arthrodesis is performed because it usually has a good prognosis, particularly in a young horse [20].

The treatment recommendations for OCLL vary widely, but the aim of all treatments is the same: to minimize the size of the cyst, promote healing of any defect within the cartilage or fibrocartilage, and increase the quality and quantity of subchondral bone present at the site of the lesion. However, if an athletic outcome is desired, then generally, surgical intervention for SCL treatment in any location is recommended [23]. In the literature, a number of surgical treatments have been published. From 1975 to 1978, cysts in the joint were approached extra-articularly by drilling the proximal phalanx from the lateral or medial side, and the defect was packed with cancellous bone graft [24]. An intra-articular procedure was used with arthroscopy of the medial femorotibial joint between the middle and medial patellar ligaments, with the joint flexed, from 1979 to 1998 [25]. Early research used an arthrotomy technique for curettage, with excellent results (42 of 51 horses), but this rate was never repeated, and arthrotomy approaches were abandoned. Arthroscopy allows direct debridement of SCLs of the MFC and basically of all lesions that are arthroscopically accessible. Otherwise, most of the SCLs of the distal limb is not accessible arthroscopically and must be debrided transcortically. Surgical debridement is now done either intraarticularly under arthroscopic supervision or transcortically. However, this method requires careful planning when drilling the cyst through the bone using digital radiography, fluoroscopy, or CT guidance.

Surgical debridement of OCLLs is recommended for horses who do not respond to conservative treatment [17,26,27]. OCLL cavities may be left open after surgical debridement or filled with cancellous bone graft [24,26], tricalcium phosphate granules [28], multiple osteochondral autografts in the form of a mosaicplasty [29], allogenic chondrocytes and IGF-I [27], or parathyroid hormone peptide (PTH_1–34_) [30]. Other treatment alternatives for OCLLs include corticosteroid injections directly into the OCLL [31] and bisphosphonates. Recently, it was reported that a transcondylar screw placed in lag fashion across SCLs of the MFC appeared a promising treatment option in managing the condition. This study found that 75% of cases had resolution of lameness and an increase in bone density of the SCL on radiographs, suggesting that cyst cavity filled with new bone [32]. In another study, a bone screw inserted into SCLs in the medial aspect of the proximal radius reduced or eliminated lameness in 7 of 8 horses (88%) [33]. Our CPB’s successful use in experimentally-induced articular cartilage defects in 15 pigs has been published previously [34]. External coaptation or surgical arthrodesis are two options for managing pathological conditions in the PIPJ. Long-term casting is often successful in metacarpophalangeal luxations/subluxations, but it is rarely so in the PIPJ, resulting in continued palmar/plantar subluxation [35]. Pressure sores and metacarpophalangeal joint laxity are possible side effects of cast immobilization in a young foal. Transarticular lag screw fixation in either parallel or cruciate fashion, dorsal plate (s) fixation, or a combination of the two are surgical choices for PIPJ arthrodesis [5]. We selected crossed, transarticular lag screw placement for the foal in this case report to increase PIPJ dorsal stability, promote bone remodeling in the cyst with CPB, and stabilize potential fissure lines that can be a cause of OCLL. In this case, however, we presumed that repairing the limb with 4.5 mm cortical self-tapping bone screws in a crossed pattern would be strong enough to support the filly in a weight-bearing position, with the limb originally supported by casting and later by a heavy bandage. Placement of the axial screw from P2 to P1 is a critical aspect considering the dorsal recess of the distal interphalangeal joint and coronary band with hoof capsule. All these structures are limitations for this type of arthrodesis, but careful radiological and surgical planning can help us to minimalize the affecting of these structures. From our previous knowledge, we conclude that the use only of 4.5 mm screws, either in a parallel or modified crossed method, is possible in those cases where the palmar soft structures (ligaments and tendons) are not disrupted or relaxed. Otherwise, successful surgical arthrodesis of PIPJ using two 5.5 mm cortical screws placed in parallel, lag fashion in a one-month-old foal with SDFT and straight sesamoidean ligament disruption has been published. Because the correct drilling of the glide hole and the threaded hole for the axial screw is crucial for successful arthrodesis, we were prepared to place a 5.5 mm cortical screw in parallel lag fashion in case of incorrect drilling. Otherwise, our clinical experiences prove that the use of the crossed method of arthrodesis with 4.5 mm screws in horses weighing 400 kg and more seems to be unstable (axial screw bending and breakage) even after the limb is fixed in the cast for a long time after the surgery. It all depends on the traumatic conditions located in the pastern region and on the weight and age of the horse. All of this information needs to be verified and updated in clinical and experimental studies.

## 5. Conclusions

This case report describes the successful return to athletic function of a filly following treatment of an OCLL in the distal aspect of the proximal phalanx using arthrodesis of the proximal interphalangeal joint with three cortical bone screws positioned in a crossed lag fashion in conjunction with intra-lesional CPB. The location of the lesion in this case allowed direct access to the cystic cavity for intra-articular curettage, filling with cement, and stabilizing by cortical screws. To our knowledge, this is the first published report of an OCLL being treated in this location with this material and arthrodesis technique. It is important to note here that the OCLL was localized between the planned position of the axial and abaxial screw, which means that the technique of insertion of an axial screw from the proximal aspect of P2 to the plantarodistal aspect of P1 does require careful surgical radiographic planning. Therefore, we tried to prevent the cyst from being hit by the screws to ensure better stabilization of the arthrodesis construct. In this case, bone remodeling in the OCLL was stimulated by the CPB. The insertion of this axial lag screw from the dorsal part of P2 to the palmar/plantar part of P1 in a dorsodistal–plantoproximal direction resulted in a stronger PIPJ arthrodesis. The third screw provides fixation and compression effects on the joint surfaces perpendicular to the effect of the historically used dorsoproximal–plantodistal screws. In contrast to the simple third lag screw fixation, the use of a plate in foals, especially at very young ages, is far too robust [6]. In young foals, application of this screw is possible because of better exposition of the pastern bones from the joint capsule. In horses older than 12–18 months, the use of this screw will be limited due to collision with the hoof capsule. Nowadays, our efforts are focused on the experimental study, where we compared the biomechanical characteristics of two methods for PIPJ arthrodesis, a conventional method with three parallel transarticular cortical screws inserted in lag fashion and modified, crossed method (as in this manuscript). In both methods were used 4.5 mm cortical self-tapping screws. The biomechanical study consists of mechanical testing of cadaveric limbs in three-point bending in a dorsal-to-plantar/palmar direction using a material testing machine. Experimental measurements show that in five of the seven cases (cadaveric limbs), the strength of the crossed arthrodesis method was greater compared to the conventional method. A significant difference in maximum bending and strength was noted in three cases, where the crossed method of arthrodesis withstood two to three times greater load under the same loading (3-point bending) (unpublished results). These results may support our view that this method of arthrodesis provides better dorso-palmar/plantar stabilization of PIPJ.

In our case, success was understood as the absence of lameness and a return to previous activity levels. Resolution of lameness was seen at 12 weeks postoperatively. A limitation of this study, however, is that the data were presented for a single case, and no controls were used to fully evaluate treatment options. Further research is required to assess the effectiveness of biological products alone and in conjunction with transcystic screw placement before a decision on their efficacy can be made.

In summary, this report demonstrates the effective management of a filly following treatment of an articular OCLL in the distal, proximal phalanx by the placement of a cortical bone screw in a crossed lag fashion and application of CPB into the cyst. This resulted in total resolution of lameness at 12 weeks post-surgery and almost total absence of radiographic evidence of the OCLL by 14 weeks post-surgery. Given the successful outcome, in this case, future research is required to further assess this surgical technique and CPB application on a larger number of cases in this and other locations for OCLL formation in the horse.

## Figures and Tables

**Figure 1 animals-11-00948-f001:**
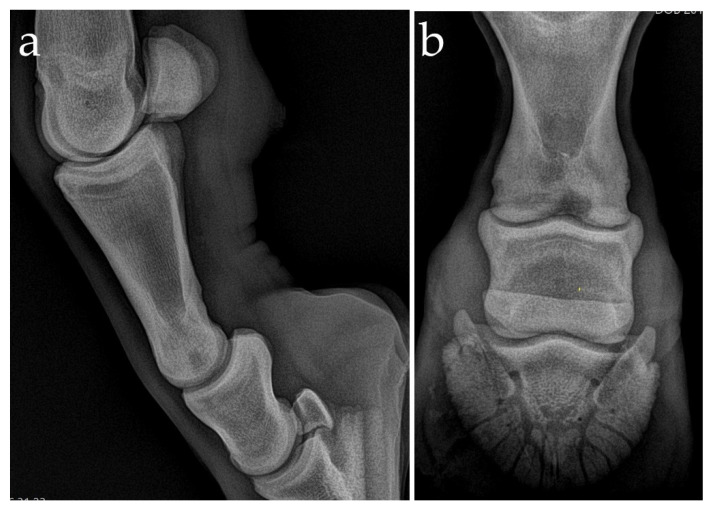
Lateromedial (**a**) and dorsoproximal–plantarodistal oblique (**b**) radiograph of the proximal interphalangeal joint of the left hindlimb. There is an ill-defined sagittal radiolucency within the distal subchondral bone of the proximal phalanx. Wide communication between the proximal interphalangeal joint and the osseous cyst-like lesions (OCLL) is clear on these radiographs.

**Figure 2 animals-11-00948-f002:**
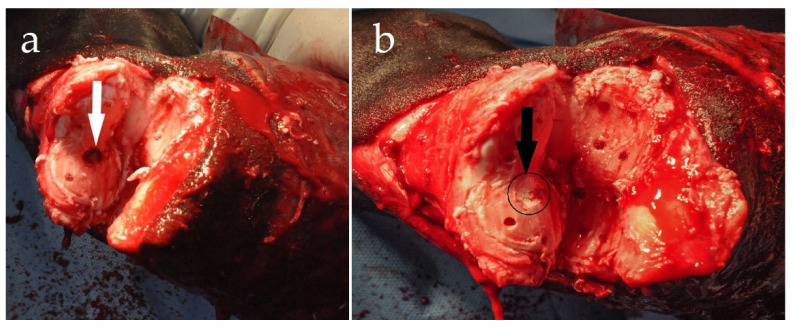
Complete cartilage removal after disarticulation of the proximal interphalangeal joint. (**a**) OCLL in the distal aspect of the proximal phalanx after curettage (white arrow) and (**b**) after packing with CPB (black arrow). Multiple holes were drilled through the subchondral plates (osteostixis) of the proximal and middle phalanges.

**Figure 3 animals-11-00948-f003:**
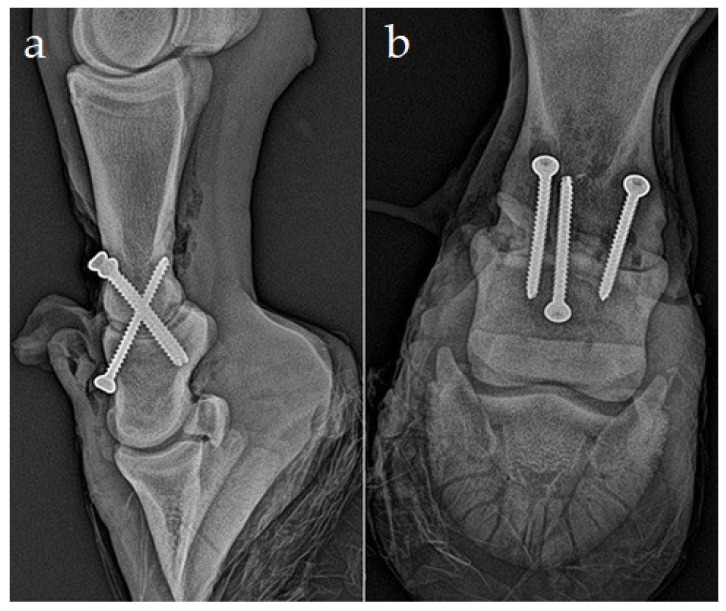
Lateromedial (**a**) and dorsoproximal–plantarodistal oblique projection (**b**) control radiographs of the proximal interphalangeal joint arthrodesis during the surgery.

**Figure 4 animals-11-00948-f004:**
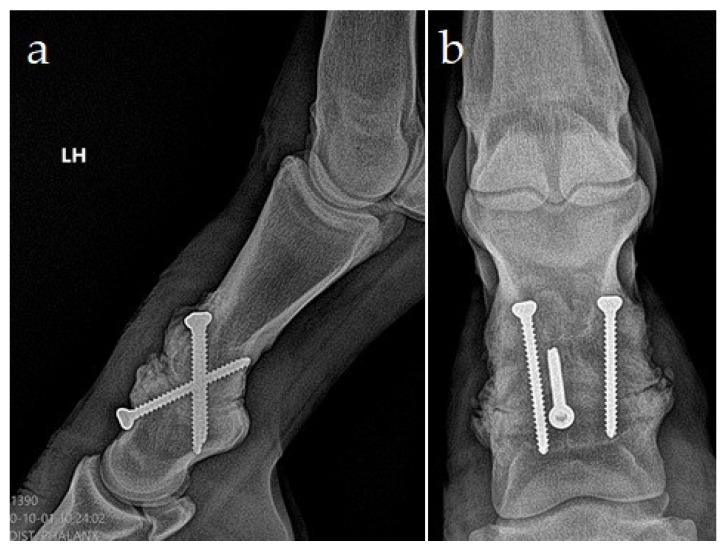
Lateromedial (**a**) and dorsoplantar (**b**) weight-bearing radiographs of the left hindlimb 3 months after surgery, demonstrating a nearly complete bony fusion of the proximal interphalangeal joint (PIPJ) and mild osseous dorsal, medial, and lateral periarticular reaction.

## Data Availability

Data are contained within the article.

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
