# Peer review of "Modified Proximal Interphalangeal Joint Arthrodesis in a Yearling Filly with an Osseous Cyst-Like Lesion in the Proximal Phalanx"

_animals, 2021, doi:10.3390/ani11040948_

Round 1

Reviewer 1 Report

General comments

- Bone cyst is not a correct term unless a histologic confirmation is available. The correct term to be used in the manuscript would be osseous cyst-like lesion. Please review the manuscript and update it

- Please change pastern to proximal interphalangeal joint

- The use of the term pastern bones is not appropriate for a scientific manuscript. Please review the manuscript

- Although the aim of this manuscript is to show the surgical technique for the treatment of the OCLL, the discussion regarding this technique and the other ones currently performed for this disease is almost inexistent. Therefore, authors should put much more emphasis on this. The beginning of the discussion and the introduction are very similar and repetitive.

- The position of the axial screw in the middle phalanx is very distal and likely affecting the dorsal recess of the distal interphalangeal joint. Please discuss this.

Specific comments

Line 14: change carpal joint to carpal bones

Line 81: use the correct terminology for radiographic projections: dorsoplantar (DPl) and lateromedial (LM)

Line 89: left hindlimb

Line 91: need to specify the joint.

Line 93: need to name the projections before using the abbreviation. The correct abbreviation would be: DPl, DLPlMO and DMPlLO

Line 93: change sagittal to axial. Provide measurements. The OCLL is ill-defined and surrounded by moderate increased opacity of the adjacent trabecular bone

Line 94: change spongiosa to trabecular bone.

Line 95: change stoma to cloaca

Figure 1: You do not mention this projection in the manuscript.

Line 100: the margination of the joint is not fully well-defined, I would suggest describing it as ill-defined.

Line 113: you use a British term (anaesthesia) but in the rest of the manuscript use the American one. Please be consistent with the language.

Line 123: Specify the collateral ligaments of which joint

Line 134: before using P1 or P2, need to state the abbreviation

Line 136: change medullar to medullary

Line 177: change dorsoplantar to dorsoproximal-plantarodistal oblique projection

Author Response

Dear Reviewer,
We would like to thank for your specific and helpful comments. We have carefully reviewed the comments and have revised the manuscript accordingly. Down below we also enclose a "Cover letter" with all the changes performed in the text of the manuscript. Changes to the manuscript are shown in yellow. As the comments of the second reviewer overlapped in part, its changes are shown in green highlight color. Please see the attachment.
We hope that you will accept our changes and our manuscript will be suitable for publication in the Special Issue of Animals journal. 

Best regards.

Author team.

Cover letter (Reviewer 1)

Title:              Pastern Joint Arthrodesis in a Yearling Filly with a Filling Subchondral Bone Cyst in the Proximal Phalanx

Author:          Filip Kolvek, Lenka Kresakova, Katarina Vdoviakova, Lubomir Medvecky, ZdenÄ›k Žert

Manuscript ID: animals-1112382

Comment 1 (from Reviewer 1)

Changes to the manuscript are shown in yellow highlight colour.

- Bone cyst is not a correct term unless a histologic confirmation is available. The correct term to be used in the manuscript would be osseous cyst-like lesion. Please review the manuscript and update it.

  • Changes are shown in lines 23, 32, 33, 36, 37, 79, 221, 227, 265, 269.

- Please change pastern to proximal interphalangeal joint.

  • This has now been amended and changes are shown in yellow highlight color (Line 2, 17, 49, 101, 208, 236, 243, 313). We suggest modifying the title of the manuscript even after the recommendation of the Reviewer 2. The proposed title: “Modified Proximal Interphalangeal Joint Arthrodesis in a Yearling Filly with an Osseous Cyst-like Lesion in the Proximal Phalanx”

- The use of the term pastern bones is not appropriate for a scientific manuscript. Please review the manuscript.

  • Changes are shown in lines 220, 221, 229.

- Although the aim of this manuscript is to show the surgical technique for the treatment of the OCLL, the discussion regarding this technique and the other ones currently performed for this disease is almost inexistent. Therefore, authors should put much more emphasis on this. The beginning of the discussion and the introduction are very similar and repetitive.

  • We have added additional information to clarify.
  • Changes are shown in lines 255-263; lines 270-276, lines 278-283, lines 285-294.

Early studies used an arthrotomy approach for curettage with good results (42 of 51 horses), but this successful rate was never being repeated and arthrotomy approaches have been abandoned. Arthroscopy allows direct debridement of SCLs of the MFC and basically of all lesions that are arthroscopically accessible. Otherwise, most of the SCLs of the distal limb are not accessible arthroscopically and have to be debrided transcortically.   Nowadays, surgical debridement is performed using either an intraarticular approach under arthroscopic supervision or a transcortical approach. However, this approach requires careful planning under digital radiography, fluoroscopy or CT guidance during drilling of the cyst through the bone.

and biophosponates. Recently it was reported that a transcondylar screw placed in lag fashion across SCLs of the MFC, appeared a promising treatment option in managing the condition. This study found that 75% of cases had resolution of lameness and an increase in bone density of the SCL on radiographs, suggesting that cyst cavity filled with new bone [32]. In another study, insertion of a bone screw into SCLs in the medial aspect of the proximal radius resulted in a substantial reduction in, or elimination of, lameness in 7 of 8 horses (88%) [33].

Treatment option for pathological conditions in PIPJ are external coaptation or surgical arthrodesis. Although long-term casting is often effective in metacarpophalangeal luxations/subluxations, it is rarely as effective in the PIPJ and results in continued palmar/plantar subluxation [35]. There is the potential for pressure sores and metacarpophalangeal joint laxity secondary to cast-immobilization in a young foal.

For the foal in this case report, we chose crossed, transarticular lag screw placement to maximize dorsal stability of the PIPJ, to improve bone remodeling in the cyst with CPB and the stabilize potential fissure lines that can be a causative factor for OCLL. However, in this case, we assumed that repair with 4,5 mm cortical self-tapping bone screws in crossed manner would be sufficiently strong to support the filly in a weightbearing position with support of the limb initially by casting, later by a heavy bandage. Placement of the axial screw from P2 to P1 is a critical aspect considering to proximity of dorsal recess of the distal interphalangeal joint and coronary band with hoof capsule. All these structures are limitation for this type of arthrodesis, but careful radiological and surgical planning can help us to minimalize affecting of these structures. Otherwise, anatomical and ageing limitations are still the subject of experimental studies. 

In the references we added a new literature:

Stashak, TS. The foot: luxation and subluxation of the proximal interphalangeal (pastern) joint. In: Adams’ Lameness in Horses, 5th ed.; Ed: Stashak, T.S., Lippincott, Williams, and Wilkins, Philadelphia, 2002; pp. 741-744.

- The position of the axial screw in the middle phalanx is very distal and likely affecting the dorsal recess of the distal interphalangeal joint. Please discuss this.

  • Discussion about this point is in lines 278-283.

Specific comments

Line 15: carpal bones

Line 91: dorsoplantar (DPl) and lateromedial (LM)

Line 99-100: left hindlimb

Line 101: We specified the joint. – proximal interphalangeal joint

Line 103: We correct abbreviation and name the projections: DPl, DLPlMO (dorsolateral-plantaromedial oblique) and DMPlLO (dorsomedial-plantarolateral oblique)

Line 105: We change sagittal to axial. Size of OCLL (12-15 mm width, 10 mm depth)

Line 106: We correct spongiosa to trabecular bone.

Line 106: change stoma to cloaca. We suggest omitting the term stoma.

Figure 1: Figure 1 is mentioned in line 107.

Line 112: We correct the margination of the cyst as ill-defined.

Line 125: We correct British term (anaesthesia) to American – anesthesia.

Line 136: We specified the collateral ligaments of the joint – proximal interphalangeal joint.

Line 148: We defined P1, P2 and added an abbreviation in brackets – proximal phalanx (P1) and middle phalanx (P2).

Line 150: We correct term medullar to medullary.

Line 190: We correct term dorsoplantar to dorsoproximal-plantarodistal oblique projection.

Reviewer 2 Report

It is an interesting case report and below display some corrections and suggestions to better it.

Proposed Title:     MODIFIED PASTERN JOINT ARTHRODESIS IN A YEARLING FILLY WITH AN OSSEOUS CYST-LIKE LESION IN THE PROXIMAL PHALANX

Line 30-31. with surgical arthrodesis, followed by packing of the SBC with calcium...

Introduction: I recommend including a short revision of the different types of surgical arthrodesis of the PIPJ

Rephrase: with surgical arthrodesis, together with the packing of the SBC with calcium....

Line 70. CPB, it is the first time in the text (apart from the abstract), so in order to easy reading, you should write in extense (calcium phosphate biocement).

Line 114-115. Please specify the constant rate infusion of medetomidine

Line 138. filling of the cavity.

Rephrase: filling the cavity

Line 145. substitute intraphalangeal by interphalangeal

Line 146. black arrow. In figure 2b the arrow is red and difficult to see. Please, put a black arrow.

Line 162. Could you specify the surgical procedure?. I understand that you put the first crossed screw disto-proximally and after that, you remove it in order to do the glide holes in the first phalanx, isn´t it?

Line 163. substitute disto-proximaly  by disto-proximally

Line 250. Do you have any information about the long-term outcome?

Line 258 Was a C-arm used during surgery?

Line 262. Substitute P3 by P1

Line 260-9. The information about the strength of this type of arthrodesis is personal and not supported by literature or biomechanical studies. The same about the use of plate for arthrodesis of the PIPJ in younger horses or the application of the third screw disto-proximally in older horses. Could you apport more information?

Which is your opinion about the use of 5.5 screws?. Perhaps with the weight of the filly, these screws would had been more appropriate.

Do you have a latero-medial radiographic image of the OCLL?. This information would be relevant in order to evaluate if one paraxial screw could pass through the cyst. In this case, the ossification of the cyst would have been stimulated by the CPB and by the screw.

Author Response

Dear Reviewer,
We would like to thank for your specific and helpful comments. We have carefully reviewed the comments and have revised the manuscript accordingly. Down below we also enclose a "Cover letter" with all the changes performed in the text of the manuscript. Changes to the manuscript are shown in green highlight color. As the comments of the second reviewer overlapped in part, its changes are shown in yellow highlight color. Please see the attachment.
We hope that you will accept our changes and our manuscript will be suitable for publication in the Special Issue of Animals journal. 

Best regards.

Author team.

Cover letter (Reviewer 2)

Title:              Pastern Joint Arthrodesis in a Yearling Filly with a Filling Subchondral Bone Cyst in the Proximal Phalanx

Author:          Filip Kolvek, Lenka Kresakova, Katarina Vdoviakova, Lubomir Medvecky, ZdenÄ›k Žert

Manuscript ID: animals-1112382

Comment 2 (from Reviewer 2)

Changes to the manuscript are shown in green highlight color. 

Proposed title: MODIFIED PASTERN JOINT ARTHRODESIS IN A YEARLING FILLY WITH AN OSSEOUS CYST-LIKE LESION IN THE PROXIMAL PHALANX

  • We accepted this title of the manuscript (Lines 2,3) with minor changes. Reviewer 1 recommends the use of a proximal interphalangeal joint and not a pastern joint. We suggest modifying the title of the manuscript even after the recommendation of the Reviewer 1. The proposed title: “Modified Proximal Interphalangeal Joint Arthrodesis in a Yearling Filly with an Osseous Cyst-like Lesion in the Proximal Phalanx”

Line 30-31. with surgical arthrodesis, followed by packing of the SBC with calcium... Rephrase: with surgical arthrodesis, together with the packing of the SBC with calcium....

  • Changes are shown in lines 30-31.

Introduction: I recommend including a short revision of the different types of surgical arthrodesis of the PIPJ ????

  • We added some information about surgical arthrodesis. Changes are shown in lines 67-74.
  • A number of techniques are recommended for PIPJ arthrodesis. The first technique routinely applied used two or three transarticular cortex screws placed in lag fashion in a parallel or diverging pattern. However, this method increases the need for cast support when compared with plating techniques. Other attempts to improve stability included the use of two plates, a T-plate, a Y-plate, and most recently a spoon plate, but currently the best clinical results have been reported by using a dorsal three-hole narrow DCP or LCP plate combined with two transarticular abaxial 5,5 mm cortical screws inserted in lag fashion [1].

Line 70. CPB, it is the first time in the text (apart from the abstract), so in order to easy reading, you should write in extense (calcium phosphate biocement).

  • Changes are shown in line 80.

Line 114-115. Please specify the constant rate infusion of medetomidine

  • This has been amended in lines 127-128.

Line 138. filling of the cavity.

  • Rephrase to filling the cavity was corrected in line 152.

Line 145. substitute intraphalangeal by interphalangeal

  • Rephrase to interphalangeal was corrected in line 156.

Line 146. black arrow. In figure 2b the arrow is red and difficult to see. Please, put a black arrow.

  • We correct Figure 2a and 2b, we highlighted and incorporated the arrows into Figure 2a and 2b (Line 154).

Line 162. Could you specify the surgical procedure?. I understand that you put the first crossed screw disto-proximally and after that, you remove it in order to do the glide holes in the first phalanx, isn´t it?

  • We specified this surgical procedure in lines 172-178. We have added additional information to clarify.
  • Firstly, we drilled the glide holes disto-proximally into proximal aspect of P2 for axial screw insertion and into distal aspect of P1 for abaxial screws insertion. After preparation of glide holes, the joint was repositioned in physiological position, the threaded hole for axial screw was drilled across the bone and followed by insertion of the 4.5 mmm cortical self-tapping screw. The thread holes for two abaxial transarticular screws were drilled and the screws were inserted and tightened.

Line 163. substitute disto-proximaly  by disto-proximally

  • We correct to disto-proximally in line 173.

Line 250. Do you have any information about the long-term outcome?

  • Some information is given in the results and conclusion (Lines 201-209). We rewrote part and added some additional information in the results section (Lines 210-213) - Ten weeks post operatively, the filly was discharged with instructions to control handwalking and then controlled training exercise. The owner reported that the filly was sound at a walk and trot. The pastern region was slighlty thickened around the dorsal aspect of PIPJ. Control radiographs were not obtained.

Line 258 Was a C-arm used during surgery?

  • We performed arthrodesis under digital radiology.
  • We correct in line 321 –surgical radiographic planning.

Line 273. Substitute P3 by P1

            We correct to P1 in line 325.

Line 260-9. The information about the strength of this type of arthrodesis is personal and not supported by literature or biomechanical studies. The same about the use of plate for arthrodesis of the PIPJ in younger horses or the application of the third screw disto-proximally in older horses. Could you apport more information? 

  • This has been amended in lines 332-344. Nowadays, our efforts are focused to the experimental study, where we compared the biomechanical characteristics of two methods for PIPJ arthrodesis, a conventional method with three parallel transarticular cortical screws inserted in lag fashion and modified, crossed method (as in this manuscript). In both methods were used 4.5 mm cortical self-tapping screws. The biomechanical study consists of mechanical testing of cadaveric limbs in 3-point bending in a dorsal-to-plantar/palmar direction using a materials testing machine. Experimental measurements show that in 5 of the 7 cases (cadaveric limbs), the strength of the crossed arthrodesis method was greater compared to the conventional A significant difference in maximum bending and strength was noted in 3 cases, where the crossed method of arthrodesis withstood two to three times greater load under the same loading (3-point bending) (unpublished results). These results may support our view that this method of arthrodesis provides better dorso-palmar/plantar stabilization of PIPJ.

Which is your opinion about the use of 5.5 screws?. Perhaps with the weight of the filly, these screws would had been more appropriate.

  • This has been amended in lines 294-308. From our previous knowledge, we conclude that the use only of 4.5 mm screws, either in a parallel or modified crossed method, is possible in those cases where the palmar soft structures (ligaments and tendons) are not disrupted or relaxed. Otherwise, successful surgical arthrodesis of PIPJ using two 5.5 mm cortical screws placed in parallel, lag fashion in a one-month-old foal with SDFT and straight sesamoidean ligament disruption has been published. Because the correct drilling of the glide hole and the threated hole for the axial screw is crucial for successful arthrodesis, we were prepared to place a 5.5 mm cortical screw in parallel lag fashion in case of incorrect drilling. Otherwise, our clinical experiences prove that the use of the crossed method of arthrodesis with 4.5 mm screws in horses weighing 400 kg and more seems to be unstable (axial screw bending and breakage) even after the limb is fixed in the cast for a long time after the surgery. It all depends on the traumatic conditions located in the pastern region and on the weight and age of the horse. All of this information needs to be verified and updated in clinical and experimental studies.

Do you have a latero-medial radiographic image of the OCLL?. This information would be relevant in order to evaluate if one paraxial screw could pass through the cyst. In this case, the ossification of the cyst would have been stimulated by the CPB and by the screw.

  • We added the latero-medial projection of the OCLL in the manuscript (Line 111).
  • We rewrote part in conclusions about screw and OCLL position in lines 318-323. It is important to note here that the OCLL was localized between the planned position of the axial and abaxial screw, which means that the technique of insertion an axial screw from the proximal aspect of P2 to the plantarodistal aspect of P1 does require careful surgical radiographic planning. Therefore, we tried to prevent the cyst from being hit by the screws to ensure better stabilization of the arthrodesis construct. In this case, bone remodeling in the OCLL was stimulated by the CPB.

Round 2

Reviewer 1 Report

Thank you for the review, I think the manuscript has been significantly improved and I have not anything else to add.